# Site-specific risk of colorectal neoplasms in patients with non-alcoholic fatty liver disease: A systematic review and meta-analysis

XiaoLi Lin[‡], FengMing You[‡], Hong Liu, Yu Fang, ShuoGuo Jin, QiaoLing Wang *

Hospital of Chengdu University of Traditional Chinese Medicine, Chengdu, Sichuan Province, China

‡ XLL and FMY contributed equally to this paper. Co-first author.
* qiaoling86@126.com

## Abstract

### Background

Many studies have shown that NAFLD is indeed closely related to the occurrence of colon tumors. The aim of this study was to further establish an assessment for the risk associated with NAFLD and the site-specificity of colon tumors.

### Methods

We searched the PubMed, Embase, Cochrane, and Scopus databases published from January 1, 1981, to December 15, 2019, assessing the risk of colorectal neoplasms in patients with NAFLD. The primary outcome measure was the incidence of site-specific risk of colorectal neoplasms in patients with NAFLD reported as ORs which pooled under a random-effects model and calculated via Mantel-Haenszel weighting. The study is registered with PROSPERO, number CRD42020162118.

### Results

11 articles (12,081 participants) were included in this meta-analysis. After heterogeneity removed, the overall risk-value pooled for right colon tumors(OR = 1.60,95% CI 1.27–2.01, I2 = 58%,P = 0.02)was higher than the left(OR = 1.39,95% CI 1.11–1.73,I2 = 59%,P = 0.02). However, this outcome was unclear when considering gender differences (Male&Right: OR = 1.05; Male&Left:OR = 1.26; Female&Right: OR = 1.17; Female&Left:OR = 1.17).The incidence of right colon tumors(Asian&Right:OR = 1.56)was obviously higher in Asians with NAFLD than the left (Asian&Left:OR = 1.23),while the risk relevance was similar and moderately associated with an increased risk of incident double-sided colorectal tumors in Europeans (European&Right:OR = 1.47; European&Left:OR = 1.41). The outcome of pathological morphology includes: the advanced adenoma OR = 1.82;the tubular adenoma OR = 1.24; the serrated adenoma OR = 2.16.

**Data Availability Statement:** All relevant data are within the manuscript and its Supporting Information files.

**Funding:** This work was supported by National Natural Science Foundation of China [No.81774284]; National Administration of Traditional Chinese Medicine : 2019 Project of building evidence based practice capacity for TCM [No. 2019XZZX-ZL006]; Xinglin Scholor Research Premotion Project of Chengdu University of TCM [No.ZRYY1909]; Science and Technology Development Fund of Hospital of Chengdu University of TCM [No.19LW10].The funders had no role in study design, data collection and analysis, decision to publish, or preparation of the manuscript.

**Competing interests:** The authors have declared that no competing interests exist.

## Conclusions

NAFLD is associated with a high risk of colon tumors, especially in regard to tumors of the right colon, which are more prevalent in Asian populations.

## Introduction

Colorectal cancer is the third most common malignancy in the world. Anatomically, the colorectal region is divided into the left colon (including the distal 1/3 transverse colon, descending colon, sigmoid colon, and rectum) and the right colon (including the cecum, ascending colon, and proximal 2/3 transverse colon), which are bounded by the splenic curvature. Neoplasms (adenoma or cancer) at different sites of colon present significant heterogeneity at the clinical, histological, and molecular levels, and could be considered as distinct tumor entities. Studies have shown that the left hemicolon has a higher incidence (61.7% vs. 33.5%) of tumors, most of which occur in males (61.4%). Tumors of the right colon are more common in elderly and female patients (51.2%) [1–3]. Even though the right hemicolon has a lower incidence of tumors, their occurrence is more insidious and patients' compliance to chemotherapy and targeted drugs is worse, promoting a lower median OS (Overall Survival) that should not be ignored. The occurrence of such tumors is closely related to a higher CIMP+ and BRAF gene mutation rate and stage dependence. The left/right prognosis for stage I is similar, with the prognosis for right colon cancer better for stage II but significantly worse compared with the left colon for stage III/IV (stage III: HR = 1.06, 95% CI: 1.02 ~ 1.11) (stage IV: HR = 1.22, 95% CI: 1.16 ~ 1.28) [4–6]. As such, this study actively sought independent risk factors for early screening and intervention for right and left hemicolon tumors, especially those of the right hemicolon, while also exploring specific treatment options.

Non-alcoholic fatty liver disease (NAFLD) is composed of a variety of metabolic (dysfunction) associated fatty liver disease, including non-alcoholic steatohepatitis (NASH), fibrosis, simple steatosis, cirrhosis, and hepatocellular carcinoma with low alcohol consumption (<30 g/day for men, <20 g/day for women). Hypertension, T2DM, hyperlipidemia and other metabolic syndromes, insulin resistance (IR), and obesity are important risk factors shared by the occurrence and progression of colon tumors and NAFLD. Lifestyle and diet factors, instead of the traditional genetic syndromes (such as Gardner syndrome, genetic nonpolyposis, and inflammatory bowel disease), are the most important risk factors for colon tumors, and these have become important new research topics. Although the pathological mechanisms of NAFLD are not clear, many studies have shown that NAFLD is closely related to the occurrence of colon tumors.

In 1998, the concept of a "hepato-intestinal axis" was proposed by Marshall, suggesting that the liver and intestinal tract maintain a similar anatomical structure and possess an acquired functional relationship to some extent. In patients with NAFLD, the increased levels of insulin-related factors, including insulin-like growth factor (IGF-1) and pro-inflammatory cytokines (TNF-$\alpha$, IL-6, and IL-8), provide an ideal microenvironment for the growth of colonic mucosal tumors. Lower adipokines and higher leptin levels might also mediate NAFLD and carcinogenesis. Adipokines regulate metabolism, inflammation, and fibrogenesis and can inhibit colon tumor growth via cyclic AMP-activated protein kinases, inducing caspase-dependent pathways, which are involved in endothelial cell apoptosis [7–11]. In addition, a large number of clinical observations have confirmed the relationship between NAFLD and colon neoplasms and its association with specific tumor locations. This current study searched the

PubMed database and found four related systematic evaluations, all of which demonstrated a high-risk correlation between NAFLD and colon tumors [12–15]. Unfortunately, only two articles included subgroup analysis for specific locations, and only two and four studies were included due to their premature publication year (2014 and 2015, respectively). Additionally, many high-quality studies published in the last five years were also not included. This current study aimed to establish an assessment of the risk associated with NAFLD and the site-specificity of colon tumors while also exploring the influence of gender, nationality, and polyp pathological morphology factors with the goal of promoting the early screening and prevention of colon tumors.

## Methods

### Search strategy and inclusion criteria

We searched the PubMed, Embase, Cochrane, and Scopus electronic databases for English language studies published from January 1, 1981, to December 15, 2019, assessing the risk of colorectal neoplasms in patients with non-alcoholic fatty liver disease using the following search strategy: ("non-alcoholic fatty liver disease" or "fatty liver" or "non-alcoholic steatohepatitis" or "NAFLD") AND ("colorectal neoplasms" or "colorectal neoplasm" or "colorectal adenomas" or "colorectal adenoma" or "colorectal cancer" or "colorectal tumor" or "colorectal carcinoma" or "colonic neoplasms" or "colonic neoplasm" or "colonic adenomas" or "colonic adenoma" or "colonic cancer" or "colonic tumor" or "colonic carcinoma" or "colon cancer" or "adenomatous polyps").

Inclusion criteria were (a) published as an original article; (b) used a cohort or cross-sectional design; (c) reported the risk of colorectal neoplasms in patients with NAFLD, in terms of standardized risk as an odds ratio (OR) or relative risk (RR); (d) diagnosed by liver biopsy, imaging, or computed tomography (CT) from individuals in the absence of excessive alcohol consumption and other known causes (e.g., viral, drugs) of chronic liver diseases, and the diagnosis of colon tumor was based on either biopsy or colonoscopy techniques, reporting site-specific incidence of colonic tumors in detail (left and right colon or proximal colon and distal colon or sigmoid, transverse, descending, ascending colon, splenic flexure, rectum, and cecum). Exclusion of selected studies from our meta-analysis included: a) NAFLD diagnosis based exclusively on serum liver enzymes; b) publication time before the 1980s; c) Result data were incomplete or had significant publication duplicates. No restrictions regarding age, sex, race, complications, duration, or location of the study were applied.

### Data extraction

Two authors (Wang and Lin) independently extracted the following data from all of the eligible studies using a predefined data extraction form: study characteristics (authors, year, and study design), study setting (region), study population characteristics (sample size and other basic data), outcomes (percentage of NAFLD and no-NAFLD patients with left or right colonic tumors), and detailed adjusted factors for effect index. The Ottawa scale (NOS) [16] was used as a quality evaluation tool, and selection, comparability, and outcome were gauged by assigning points based on the NOS values from 1 to 9. Studies with a rating of 6 or higher were considered of high quality [17]. Areas of discrepancies or uncertainty were resolved by consensus.

### Data synthesis and analysis

The primary outcome measure was the incidence of the site-specific risk of colorectal neoplasms in patients with NAFLD, reported as ORs and pooled under a random-effects model

[18]. The OR for each study was defined as the reported number of NAFLD patients diagnosed with left or right colorectal neoplasms compared with the number of no-NAFLD patients. The 95% confidence intervals were considered as the effect size for all of the eligible studies. A random-effects analysis method was also used to combine studies into defined subgroups based on gender, pathologic tumor morphology, study design type, or nationality, thus assigning a relative weight to each study within the subgroup that summed to 100%. The relative weight of each study was calculated via Mantel-Haenszel weighting [19].

We assessed heterogeneity across the studies using the $I^2$ statistic and Cochran's Q statistic. $I^2<50\%$ and $P>0.10$ indicated a substantial heterogeneity. The heterogeneity of the studies was minimized by adjusting the form of the effect model (e.g., replace the fixed effects model with a random-effects model) or the selection of the effect size (e.g., replace OR with RR) if there was obvious heterogeneity. Sensitivity analysis of each individual study in the pooled analysis results and subgroup influence analysis was performed, if necessary. The possibility of small-study effects (publication bias) was assessed across the studies using a funnel plot as well as Egger's and Begg's regression asymmetry tests.

This systematic review and meta-analysis were performed according to prespecified criteria and followed the PRISMA [20] and MOOSE [21] guidelines for the reporting of meta-analyses with Review Manager 5.1. The study is registered with PROSPERO, number CRD42020162118.

## Results

### Characteristics of the studies

The literature search identified 2554 records, of which 2522 were excluded after duplicates were removed and an initial screening of titles and abstracts. A total of 32 full-text articles were assessed for eligibility, and all of the studies that did not mention specific sites of the colon tumors or report complete dates were excluded during this process. Finally, only 11 articles [8, 13, 22–30] were included in the meta-analysis after further screening and discussion (Fig 1). In two of these articles [22, 23], sex was restricted (one to male and the other to female), and these were excluded from analyses of the overall risk relationship between site-specific risk of colorectal neoplasms in patients with NAFLD, but they were included in analyses of the incidence ratio based on gender factors in their respective subgroups. Table 1 shows the overall study characteristics of these 11 studies, including first author, publication year, study design, region, sample size, outcomes regarding the percentage of NAFLD and no-NAFLD patients with left or right colonic tumors, NOS score, and other basic data. The median Newcastle-Ottawa rating for the six studies included was 6.5, of which only one study received a quality rating below the defined high-quality research standard of 6 stars. There was no inter-rater disagreement for either the screened or extracted data.

A total of 4 of the 11 studies reported the risk correlation relationship in regard to the gender specificity of patients with NAFLD suffering from left- or right-half colon tumors, with each gender involving three studies. From the pathological tumor morphology, four papers presented advanced tumors, two papers included tubular adenoma, and three papers included serrated adenoma. In terms of the study design type, there were four cross-sectional studies and five studies classified as longitudinal studies. The geographical distribution of these studies was relatively balanced, with five of them carried out in Asia and four studies carried out in Europe. Each of these classifications was included in the subgroup analyses.

The overall pooled risk value of the right and left colon tumors.

The analyzed data indicated that the significant heterogeneity between NAFLD and risk of incident colorectal tumors was consistent for both left (OR = 1.32, 95% CI 1.05–1.65, $I^2$ = 62%,

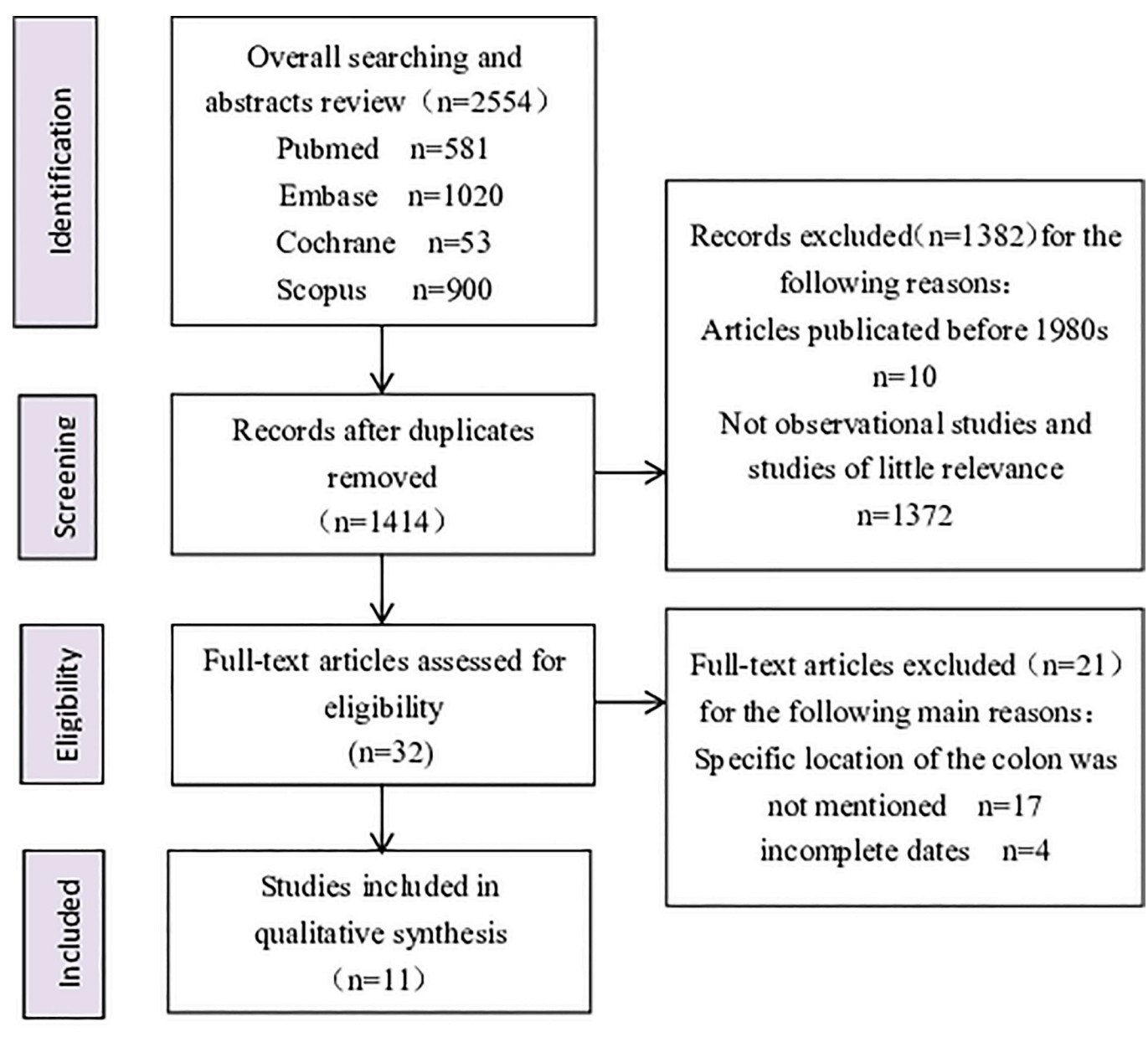

**Fig 1. Flow diagram.**

$P$ = 0.006) and right colonic tumors (OR = 1.47, 95% CI 1.14–1.90; $I^2$ = 71%, $P$ = 0.0006). Although, visual inspection of the funnel plot showed no major asymmetrical appearance (Fig 2), and no small-study effects or publication bias were apparent as assessed via Begg's test (p = 0.139) and Egger's test ($P$ = 0.203); however, the reliability of these results could not be ensured. As such, stratified analyses, meta-regression, and sensitivity analyses were performed. The effect size and effect models were varied and debugged, and an influence test that eliminated each of the included studies one at a time was used to explore for possible sources of heterogeneity across the studies. Fig 3 shows the final results after the heterogeneity reduction. The overall pooled risk value of the right colon tumors among the nine studies (OR = 1.60,

**Table 1. Characteristics of studies in meta-analysis.**

| Authors Years | Region | Design | Size | NAFLD | | No-NAFLD | | Basic data | NOS Score |
|---|---|---|---|---|---|---|---|---|---|
| | | | | Right | Left | Right | Left | | |
| Sang Tae Hwang, 2010 | Kangbuk Korea | Cross-sectional | 2917 | 13.54% | 10.90% | 8.22% | 9.01% | Gender, age, BMI, waist, HBP, FPG, HDL, LDL, triglyceride, GGT, HOMA-IR, AST, ALT, DM, Hypertensionsm, Diabetes, smoking | 6 |
| Stadlmayr 2011 | Oberndorf Austria | cohort | 1211 | 25.32% | 28.01% | 16.93% | 15.03% | age, BMI, Waist, HBP, TG, LDL, HDL, Triglycerides, Uricacid, Fastingglucose, Fastinginsulin, OGTT1, OGTT2, HOMA-IR, HbA1c, GGT, AST, ALT, Erythrocytesedimentationrate, CRP, Haemoglobin | 6 |
| Nadege T. Touzin 2011 | San Antonio USA | cohort | 233 | 26.60% | 22.34% | 25.90% | 28.06% | Gender, age, BMI, Ethnicity, HDL, LDL, triglyceride, GGT, HOMA-IR, AST, ALT, DM, Hypertensionsm, Diabetes, smoking, NAFLD | 6 |
| Huafeng Shen 2013 | New York USA | Cross-sectional | 580 | 13.42% | 12.15% | 10.81% | 10.27% | Gender, age, BMI, Race, Tobacco use, Alcohol use, Diabetes, Hypertension, Dyslipidemia, Family history of colorectal cancer | 5 |
| K.-W. Huang 2013 | Taipei China | cohort | 1522 | 11.29% | 8.06% | 5.21% | 5.43% | gender, Age, Follow-up, BMI, Waist, Waist-to-height ratio, Fasting glucose, ALT, AST, Cholesterol, HDL, LDL, Triglyceride, Hypertension, Diabetes, Smoking, Metabolic syndrome | 7 |
| Vincent Wai-Sun Wong 2014 | HongKong China | cohort | 380 | 23.62% | 18.59% | 8.84% | 13.81% | Age, years, Gender, Ever smoker, Colorectal cancer in first degree relatives, BMI, Waist, Fasting glucose, Total cholesterol, HDL, LDL, Triglycerides, ALT, AST, Diabetes, Hypertension, Hepatic triglyceride content, Steatosis grade, Lobular inflammation, Ballooning, Fibrosis stage | 7 |
| Qin-Fen Chen 2017 | WenZhou China | Cross-sectional | 2409 | 11.25% | 20.42% | 11.53% | 17.85% | Gender, Age, Weight, Height, BMI, SBP, DBP, FPG, TG, TC, HDL-C, LDL-C, ALT, AST, NAFLD, MS, Smoking, Alcohol | 7 |
| Young Joo Yang 2017 | Chuncheon Korea | cohort | 1023 | 10.66% | 10.88% | 12.54% | 10.48% | Age at diagnosis (years), BMI, SBP, DBP, Triglycerides, HDL, LDL, Fasting glucose, CEA, MetS | 7 |
| Xiao-Jun YU 2018 | ShangHai China | Cross-sectional | 673 | 12.21% | 36.01% | 12.90% | 56.69% | Gender, age, BMI | 6 |
| John William Blackett 2019 | New York USA | Cross-sectional | 369 | 21.14% | 10.57% | 15.85% | 6.10% | Age, Gender, Surveillance, Metabolic syndrome comorbidities, Hyperlipidemia, Obesity, Diabetes | 8 |
| Zhou-Feng Chen 2018 | WenZhou China | Cross-sectional | 764 | 24.05% | 75.95% | 27.68% | 72.32% | Age, BMI, SBP, DBP, Triglycerides, HDL, LDL, Fasting glucose, CEA, Differentiation | 7 |

95% CI 1.27–2.01, $I^2$ = 58%, $P$ = 0.02) was higher in patients with NAFLD than in the general population; the same conclusion could also be applied to the left colon tumors with a slightly lower risk value (OR = 1.39, 95% CI 1.11–1.73, $I^2$ = 59%, $P$ = 0.02).

After adjustment for factors that contribute to heterogeneity, these results were still significant, although the significance was decreased to some extent compared to that in the crude results. It was considered that this change might be related to gender, ethnic disparity, study design, and/or another variable; therefore, a series of subgroup analyses were performed to explore this possibility.

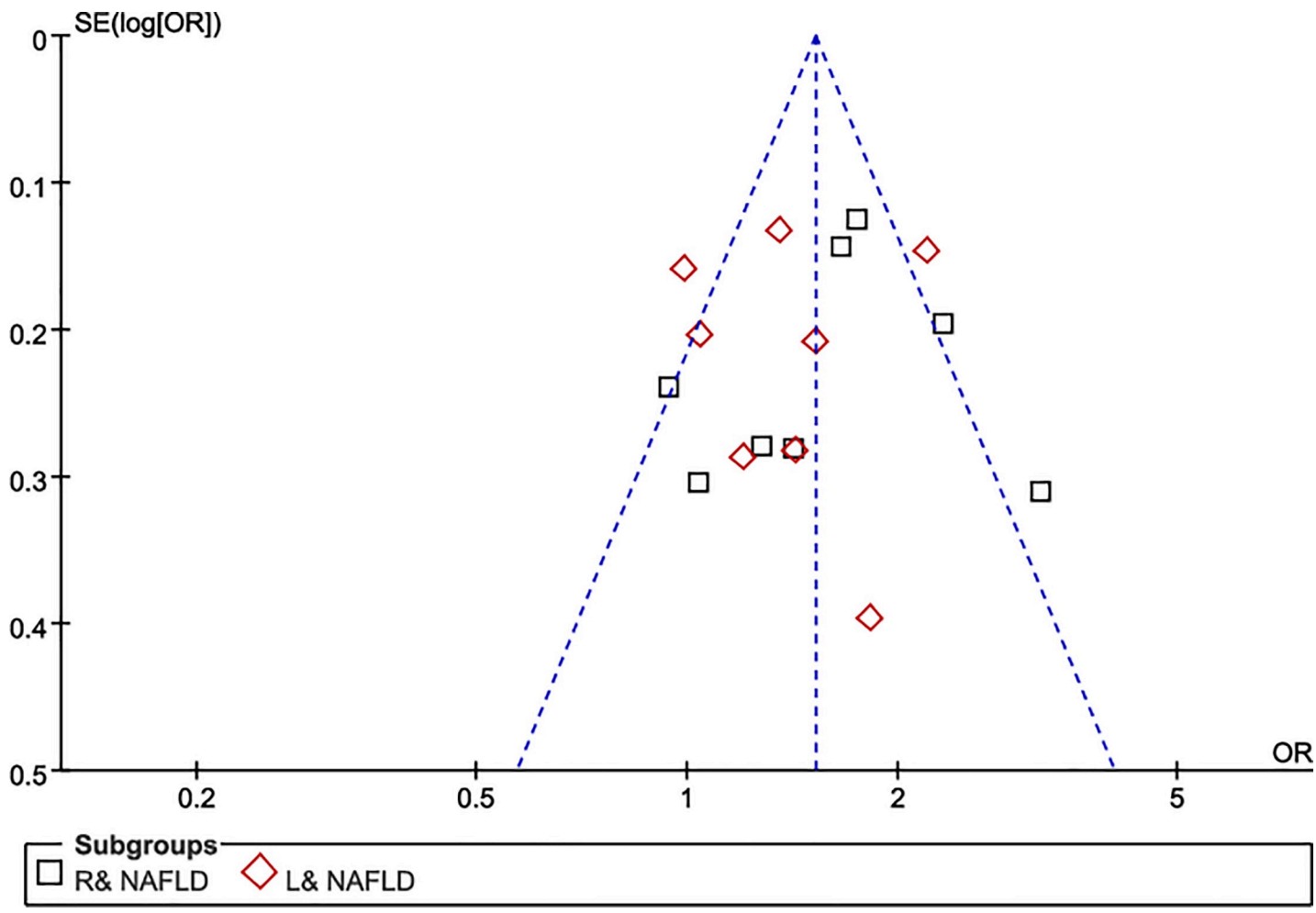

**Fig 2. Funnel plot.**

## Subgroup of study design

The types of study design used were also distinguished. The results of the five risk-association studies attributed to cohort studies showed that the incidence of right colon tumors in patients with NAFLD was significantly higher than that of left tumors (Cohort study & R: OR = 1.59, 95% CI 1.02–2.46, $I^2$ = 81%, $P$ = 0.0003; Cohort study & Left: OR = 1.35, 95% CI 0.93–1.97, $I^2$ = 74%, $P$ = 0.004). The results of the other four risk-association studies attributed to the cross-sectional studies showed a slightly higher risk in the right colon (Cross-sectional study & R: OR = 1.37, 95% CI 1.02–1.85, $I^2$ = 47%, $P$ = 0.13; Cross-sectional study & Left: OR = 1.22, 95% CI 1.00–1.50, $I^2$ = 11%, $P$ = 0.34) (Table 2). These data were consistent with the conclusion presented above.

## Subgroup of gender

As shown in Table 2, NAFLD was significantly associated with the number of left or right colorectal tumors, as indicated above. However, it was unclear in regard to sex difference (Male & Right: OR = 1.05, 95% CI 0.85–1.32, $I^2$ = 2%, $P$ = 0.36; Male & Left: OR = 1.26, 95% CI 1.05–1.51, $I^2$ = 0%, $P$ = 0.38; Female & Right: OR = 1.17, 95% CI 0.70–1.96, $I^2$ = 73%, $P$ = 0.02;

| Study or Subgroup | NAFLD Events | Total | non-NAFLD Events | Total | Weight | Odds Ratio M-H. Fixed. 95% CI | Odds Ratio M-H. Fixed. 95% CI |
|---|---|---|---|---|---|---|---|
| **1.1.1 R& NAFLD** | | | | | | | |
| A. Stadlmayr2011 | 160 | 632 | 98 | 579 | 11.1% | 1.66 [1.26, 2.21] | |
| Huafeng Shen2013 | 53 | 395 | 20 | 185 | 3.4% | 1.28 [0.74, 2.21] | |
| John William Blackett2019 | 26 | 123 | 39 | 246 | 3.0% | 1.42 [0.82, 2.47] | |
| K.-W. Huang2013 | 70 | 620 | 47 | 902 | 4.9% | 2.32 [1.58, 3.40] | |
| Nadege T. Touzin2011 | 25 | 94 | 36 | 139 | 3.1% | 1.04 [0.57, 1.88] | |
| Sang Tae Hwang2010 | 128 | 945 | 162 | 1972 | 13.1% | 1.75 [1.37, 2.24] | |
| Vincent Wai-Sun Wong,2014 | 47 | 199 | 16 | 181 | 1.9% | 3.19 [1.74, 5.86] | |
| Xiao-Jun YU2018 | 32 | 262 | 53 | 411 | 5.2% | 0.94 [0.59, 1.50] | |
| Young Joo Yang2017 | 47 | 441 | 73 | 582 | 0.0% | 0.83 [0.56, 1.23] | |
| Subtotal (95% CI) | | 3270 | | 4615 | 45.7% | 1.65 [1.44, 1.89] | |
| Total events | 541 | | 471 | | | | |
| Heterogeneity: Chi² = 16.70, df = 7 (P = 0.02); I² = 58% | | | | | | | |
| Test for overall effect: Z = 7.17 (P < 0.00001) | | | | | | | |
| | | | | | | | |
| **1.1.2 L& NAFLD** | | | | | | | |
| A. Stadlmayr2011 | 177 | 632 | 87 | 579 | 9.5% | 2.20 [1.65, 2.93] | |
| Huafeng Shen2013 | 48 | 395 | 19 | 185 | 3.3% | 1.21 [0.69, 2.12] | |
| John William Blackett2019 | 13 | 123 | 15 | 246 | 1.3% | 1.82 [0.84, 3.96] | |
| K.-W. Huang2013 | 50 | 620 | 49 | 902 | 5.3% | 1.53 [1.02, 2.30] | |
| Nadege T. Touzin2011 | 21 | 94 | 39 | 139 | 0.0% | 0.74 [0.40, 1.36] | |
| Sang Tae Hwang2010 | 103 | 945 | 163 | 1972 | 13.6% | 1.36 [1.05, 1.76] | |
| Vincent Wai-Sun Wong,2014 | 37 | 199 | 25 | 181 | 3.1% | 1.43 [0.82, 2.48] | |
| Xiao-Jun YU2018 | 148 | 262 | 233 | 411 | 11.4% | 0.99 [0.73, 1.36] | |
| Young Joo Yang2017 | 48 | 441 | 61 | 582 | 6.8% | 1.04 [0.70, 1.56] | |
| Subtotal (95% CI) | | 3617 | | 5058 | 54.3% | 1.41 [1.24, 1.61] | |
| Total events | 624 | | 652 | | | | |
| Heterogeneity: Chi² = 17.24, df = 7 (P = 0.02); I² = 59% | | | | | | | |
| Test for overall effect: Z = 5.15 (P < 0.00001) | | | | | | | |
| | | | | | | | |
| Total (95% CI) | | 6887 | | 9673 | 100.0% | 1.52 [1.38, 1.67] | |
| Total events | 1165 | | 1123 | | | | |
| Heterogeneity: Chi² = 36.60, df = 15 (P = 0.001); I² = 59% | | | | | | | |
| Test for overall effect: Z = 8.68 (P < 0.00001) | | | | | | | |
| Test for subgroup differences: Chi² = 2.65. df = 1 (P = 0.10). I² = 62.2% | | | | | | | |

**Fig 3. Forest plot.**

Female & Left: OR = 1.17, 95% CI 0.79–1.72, $I^2$ = 58%, P = 0.09). It should be noted that the number of included studies (n = 4) was relatively small, and an increased number of studies will be needed to confirm this finding. (Note: "Male&Right" means the risk correlation of the right colon tumor between male and NAFLD."Male&Left" means the risk correlation of the left colon tumor between male and NAFLD.The following"Female&Right"""Female&left""Asian&Right""Asian&Left""European&Right""European&Left" keep the same interpretive mode as above.)

## Subgroup of nationality

Subsequently, the effects of ethnic disparity, such as different lifestyle habits, body fat distribution, and cultural background, were analyzed relative to the risk correlation of site-specific colorectal neoplasms among the patients with NAFLD. The incidence of right colon tumors

**Table 2. Sensitivity and subgroup analyses about the association between NAFLD and the risk of right and left-side colorectal neoplasm.**

|  | Right colorectal neoplasm | Left colorectal neoplasm |
|---|---|---|
| **Study design type** | | |
| **Cohort study** | OR = 1.59, 95%CI 1.02–2.46 | OR = 1.35, 95%CI 0.93–1.97 |
|  | $I^2$ = 81%, p = 0.0003, N = 5 | $I^2$ = 74%, p = 0.004, N = 5 |
| **Cross-sectional** | OR = 1.37, 95%CI 1.02–1.85 | OR = 1.22, 95%CI 1.00–1.50 |
|  | $I^2$ = 47%, p = 0.13, N = 4 | $I^2$ = 11%, p = 0.34, N = 4 |
| **Gender** | | |
| **Male** | OR = 1.05, 95%CI 0.85–1.32 | OR = 1.26, 95%CI 1.05–1.51 |
|  | $I^2$ = 2%, p = 0.36, N = 3 | $I^2$ = 0%, p = 0.38, N = 3 |
| **Female** | OR = 1.17, 95%CI 0.70–1.96 | OR = 1.17, 95%CI 0.79–1.72 |
|  | $I^2$ = 73%, p = 0.02, N = 3 | $I^2$ = 58%, p = 0.09, N = 3 |
| **Nationality** | | |
| **Asian** | OR = 1.56, 95%CI 1.01–2.42 | OR = 1.23, 95%CI 1.04–1.45 |
|  | $I^2$ = 84%, p<0.0001, N = 5 | $I^2$ = 9%, p = 0.36, N = 5 |
| **European** | OR = 1.47, 95%CI 1.19–1.82 | OR = 1.41, 95%CI 0.83–2.38 |
|  | $I^2$ = 0%, p = 0.51, N = 4 | $I^2$ = 74%, p = 0.008, N = 4 |
| **Tumor pathologic morphology** | | |
| **Advanced adenoma** | OR = 1.82, 95% 0.85–3.39, $I^2$ = 69%, p = 0.02 N = 4 | |
| **tubular adenoma** | OR = 1.24, 95% 0.85–1.80, $I^2$ = 0%, p = 0.62 N = 2 | |
| **serrated adenoma** | OR = 2.16, 95% 0.78–6.00, $I^2$ = 0%, p = 0.49 N = 2 | |

(Asian & Right: OR = 1.56, 95% CI 1.01–2.42, $I^2$ = 84%, $P<0.0001$) was obviously higher in Asians with NAFLD than left tumors (Asian & Left: OR = 1.23, 95% CI 1.04–1.45, $I^2$ = 9%, $P$ = 0.36), but the risk relevance was similar and moderately associated with an increased risk of incident double-sided colorectal tumors in Europeans (European & Right: OR = 1.47, 95% CI 1.19–1.82, $I^2$ = 0%, $P$ = 0.51; European & Left: OR = 1.41, 95% CI 0.83–2.38, $I^2$ = 74%, $P$ = 0.008). These data also included the problem of too few studies have been included, which might have served to decrease the robustness of these results.

## Subgroup of pathological morphology

In addition to intuitively analyzing the distribution of the tumors in the left and right colon from patients with or without NAFLD via different subgroups, the pathological morphology of the tumors could also indirectly reflect the distribution of the left and right colorectal neoplasms. The combined analysis for advanced adenoma, tubular adenoma, and serrated adenoma revealed that the pooled incidence rate (OR) for advanced adenoma was 1.82 (95% CI 0.85–3.39, $I^2$ = 69%, $P$ = 0.02), for tubular adenoma it was 1.24 (95% CI 0.85–1.80, $I^2$ = 0%, $P$ = 0.62), and for serrated adenoma it was 2.16 (95% CI 0.78–6.00, $I^2$ = 0%, $P$ = 0.49).

## Discussion

No formal guidelines or recommendations regarding routine cancer screening for colorectal neoplasms among patients with NAFLD have been established at this point, even though a substantial proportion of clinical observations and system analyses in this field have demonstrated their close correlation. The analyses in this current study have further clarified the risk of site-specific colorectal neoplasms in these patients, and the findings of this meta-analysis have provided further evidence to support the idea that the risk of colorectal neoplasms is increased among patients with NAFLD, especially tumors the stem from the right-side colon

(OR = 1.60). In view of the relatively few studies included in this analysis (n = 11), it was hard to define the site-specific risk correlation between colorectal tumors and NAFLD. Instead, the specific pathological morphologies of the tumors were assessed, including advanced adenoma, tubular adenoma, and serrated adenoma. It is noteworthy that there were a significantly higher prevalence and incidence of advanced adenoma (OR = 1.82) and serrated adenoma (OR = 2.16) relative to the general population, which is consistent with the conclusion presented above. Tubular adenoma, more common in the left colon, presented at a lower detection rate in this analysis.

Previous studies have shown that, in slightly older women, right-sided colon cancer is more common and presents more often as advanced (T3/T4) tumors; in contrast, left-side colon cancer is less common in this population. The results in this current study did not clearly define this gender difference. To clarify this difference, a comprehensive search of studies that observed the effect of NAFLD on the risk of site-specific colorectal tumors based on sex-specificity was performed in case there were any possible omissions. One study limited to men and one study limited to women within our predefined inclusion criteria could have been included [22, 23]. However, the generalizability of these findings to gender expression remains uncertain. As known, differences in regional diet, cultures, and lifestyles having an effect on tumor occurrence should not be underestimated. It was apparent that the pattern observed in this analysis was more applicable to Asians compared to most people in Europe. However, a significant heterogeneity should also be taken into consideration, as Asian & R, $I^2$ = 84% and European & L, $I^2$ = 74%. It was speculated that this heterogeneity might be attributable to confounding factors that might have affected the development of colorectal tumors; however, this does not counter the fact that the number of studies included in this analysis was relatively low.

Of note, hypermutation is more prevalent in right colon cancer (RCC), which is associated with an increase in RAS and mutations in the phosphoinositide 3-kinase pathway, CpG island methylator phenotype (CIMP)-high subtypes, microsatellite instability-high subtypes, and BRAF mutations [31–33]. Therefore, RCC is likely a negative prognostic variable. The NCCN (National Comprehensive Cancer Network) guidelines recommend the use of anti-EGFR substances for the treatment of the left colon cancer (LCC) only, and many clinical trials have also confirmed that outcomes were poorer in patients with RCC who were treated with cetuximab plus FOLFIRI or FOLFOX. As such, patients with RCC might benefit more from an initial treatment with bevacizumab in combination with chemotherapy [34]. Therefore, early screening and prevention of right colon tumors appear to be of high significance.

Currently, scientists from all over the world are devoted to exploring the early signs of colon cancer, with gene mutations and exosomes becoming the major topics of research in recent years; nevertheless, specific criteria have yet to be established or generalized. No matter which screening method is adopted; however, the economic cost, safety, and convenience of the tests should be taken into consideration. We believe that patients with NAFLD could substantially benefit from more intensive or earlier colonoscopy screenings, particularly patients of Asian descent. Unfortunately, the cost-effectiveness of these recommendations still requires investigation and validation prior to implementation, and the optimum age to initiate such screening also needs further assessment [35].

The liver and gut share numerous pathophysiological pathways [36–38], and NAFLD and colorectal tumors are also intrinsically linked to each other by abnormal signals, including insulin resistance, metabolic syndrome, obesity, type-2 diabetes mellitus, and dyslipidemia. A study by Wong et al [24]. demonstrated a significantly increased incidence of advanced neoplasms in patients with NAFLD, and the majority of these patients had right-sided colon cancer. Studies from K-W. Huang [25], Sang Tae Hwang [8], and A. Stadlmayr [26] also confirm

this hypothesis; conversely, there are still some phenomena that contradict this view. The assertions that RCC more often metastasizes to the peritoneum and a greater proportion of LCCs metastasize to the liver and lungs support a closer relationship between the liver and the left colon [31]. The specific mechanism by which NAFLD is associated with the increased risk of colonic neoplasms, especially to the right colon, is uncertain, and further studies are still needed to clarify this association.

Although it is unclear whether NAFLD is directly responsible for the development of colon tumors or whether common factors of the two diseases could be responsible for the incidence, the synergistic treatment of insulin resistance, hyperlipidemia, and specific inflammation might be beneficial in delaying the progression of colon tumors. Importantly, drug regulatory agencies have not recommended specific drugs for the treatment of NAFLD. Due to its poor tolerance or adverse effects, pharmacotherapy, such as pioglitazone, obeticholic acid, GLP-1 agonists, metformin, statins and other lipid-lowering agents, and anti-hypertensive agents, which are significantly effective in alleviating serum liver enzymes and insulin resistance, have not been recommended as first-line therapies for NAFLD [38]. Although the extent to which the treatment of NAFLD affects the risk of colon tumors also needs to be further analyzed, life-style management, including weight loss and a low-salt/low-fat diet, is still considered the most effective approach for treating NAFLD and preventing colon tumors.

As mentioned above, some limitations exist in this current study. First, due to the inherent limitations of retrospective cross-sectional studies (n = 4), time-event studies are not available for further follow-up to obtain the true results regarding the possible future event incidence. Second, the distribution of the funnel plot and Begg's test are symmetrical, and the vertical coordinate representing the sample size is mostly concentrated at the top, indicating that the sample size is considerable and there is almost no publication bias. However, the heterogeneity of the results is still large, which might result in a result bias caused by the small number of studies or some other unknown issues. Third, the specific mechanism of the risk of colonic neoplasia caused by NAFLD is not yet known; specifically, the biological mechanism by which a lateral effect influences the risk of site-specific colonic neoplasms still needs to be defined. Lastly, no significant gender distribution specificity was found in the current studies; however, considering the economic cost to patients, the acceptance and safety of universal screening via colonoscopy among patients with NAFLD will be necessary to clearly define the age and gen-der targets for the initial screening.

Despite these problems, the finding of this study still has potential clinical value. A system-atic review and meta-analysis was carried out on the specific location of colon tumors in patients with NAFLD, which is beneficial to the prevention and early screening of colon tumors, especially in regard to right-side colon tumors. It is of great significance to realize an earlier clinical cure, reduce the medical burden of patients, and improve their quality of life. With this in mind, the balance of the hepato-intestinal axis should be explored from multiple perspectives. For example, whether colon cancer combined with NAFLD is associated with a high risk of liver metastasis should be assessed [39]. Additionally, the true correlation between the progression of colonic lesions and different stages of NAFLD, including NASH, fibrosis, simple steatosis, cirrhosis, and hepatocellular carcinoma [7], could provide valuable insights. Moreover, the effect of NAFLD pharmacotherapy on the risk-regulation of colon tumors rep-resents another important avenue of inquiry.

## Conclusion

In conclusion, this systematic review and meta-analysis suggest that NAFLD is associated with a high risk of colon tumors, especially tumors of the right colon, which are more prevalent in

Asian populations. These results recommend early colonoscopy screening for patients with NAFLD to reduce the prevalence and incidence of colon tumors. It is also suggested that coordinated treatment of cardiovascular, endocrine, and liver diseases should be implemented in a timely manner in patients with NAFLD. Moreover, the development of a healthy diet, exercise program, and other healthy habits is warranted.

## Supporting information

**S1 Checklist. PRISMA 2009 checklist.**
(PDF)

## Author Contributions

**Conceptualization:** FengMing You.

**Data curation:** FengMing You.

**Formal analysis:** FengMing You.

**Funding acquisition:** Hong Liu.

**Investigation:** Hong Liu.

**Methodology:** Hong Liu.

**Project administration:** Yu Fang.

**Resources:** Yu Fang.

**Software:** Yu Fang.

**Validation:** ShuoGuo Jin.

**Visualization:** ShuoGuo Jin.

**Writing – original draft:** XiaoLi Lin, QiaoLing Wang.

**Writing – review & editing:** ShuoGuo Jin, QiaoLing Wang.

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
