## [Decision Letter · Decision Letter 0]

27 Oct 2020

PONE-D-20-25326

“Site-specific risk of colorectal neoplasms in patients with non-alcoholic fatty liver disease: a systematic review and meta-analysis”

PLOS ONE

Dear Dr. Wang,

Thank you for submitting your manuscript to PLOS ONE. After careful consideration, we feel that it has merit but does not fully meet PLOS ONE’s publication criteria as it currently stands. Therefore, we invite you to submit a revised version of the manuscript that addresses the points raised during the review process.

We apologize for the delay in the review process. Peer review is under unprecedented pressure due the current COVID-19 pandemic. Your manuscript has received a positive review which will require a minor revision. The following minor comments should be addressed in your resubmission:

Line 13: Define OS the first time it is usedLine 39: typo in "adipokines"Consider mentioning in the background on NAFLD in the introduction that there is discussion that NAFLD may be better named as Metabolic (dysfunction) associated fatty liver disease.  See: https://www.metabolismjournal.com/article/S0026-0495(20)30277-8/fulltextLine 303: the statement that the specific location of colon tumors in patients with NAFLD was evaluated for the first time is overstated given the current study is a meta-analysis. Rephrase to more accurately reflect the assessment undertaken.Line 323: typo in "conflicts"

We look forward to receiving your revised manuscript.

Kind regards,

Michael W Greene, Ph.D.

Academic Editor

PLOS ONE

Journal Requirements:

2. At this time, we ask that you please provide the full search strategy and search terms for at least one database used as Supplementary Information.

3. PLOS ONE requires systematic reviews to include a detailed analysis of the quality of each study included in the review. Please attach a Supplemental file of the results of the quality assessment for each individual study assessed, broken down into individual quality assessment measures. Please also discuss how results can be interpreted given the quality of the included studies.

5. Please include a captions for figure 2 and 3.

Reviewers' comments:

Reviewer's Responses to Questions

**Comments to the Author**

1. Is the manuscript technically sound, and do the data support the conclusions?

Reviewer #1: Yes

2. Has the statistical analysis been performed appropriately and rigorously? 

Reviewer #1: Yes

3. Have the authors made all data underlying the findings in their manuscript fully available?

Reviewer #1: Yes

4. Is the manuscript presented in an intelligible fashion and written in standard English?

Reviewer #1: Yes

5. Review Comments to the Author

Reviewer #1: This is an interesting article addressing a clinically relevant issue.

6. PLOS authors have the option to publish the peer review history of their article (what does this mean?). If published, this will include your full peer review and any attached files.

Reviewer #1: No

---

## [Author Response · Author response to Decision Letter 0]

18 Nov 2020

Dear Editors: 

On behalf of my co-authors, we thank you very much for giving us an opportunity to revise our manuscript, we appreciate editor and reviewers very much for their positive and constructive 

comments and suggestions on our manuscript entitled “ Site-specific risk of colorectal neoplasms in patients with non-alcoholic fatty liver disease: a systematic review and meta-analysis” . (I D: PONE-D-20-25326 ). 

We have studied reviewer ’ s comments carefully and have made revision which marked in yellow highlight in the paper. We have tried our best to revise our manuscript according to the comments. Attached please find the revised version, which we would like to submit for your kind consideration. The following is our modification plan：

1、The English name of author affiliation has been redefined in recent months, and updated here.

2、Line 13: We defined OS the first time it is used.

3、Line 40: We respelled the word "adipokines".

4、Line 22-23: By referring to the comments of the editor, we introduced the concept of “Metabolic (dysfunction) associated fatty liver disease” into the background of NAFLD in the introduction.

5、Line 298-301: After discussion, we also agree that the term "for the first time" is not appropriate, so we rephrase it to more accurately reflect the assessment undertaken.

6、Line 324-326: dear editors, Maybe you think the content under the tittle “Acknowledgements“ should be credited in the “conflicts” title, but in Line 318-319 we have made “Conflict of interest”, so we decided to delete the part of “Acknowledgements”. 

We would like to express our great appreciation to you and reviewers for comments on our paper. 

Looking forward to hearing from you. 

Thank you and best regards. 

Yours sincerely, 

Qiao-ling Wang 

Corresponding author: 

Name: Qiao-ling Wang

E-mail:qiaoling86@126.com

---

## [Editor Report · Decision Letter 1]

11 Jan 2021

“Site-specific risk of colorectal neoplasms in patients with non-alcoholic fatty liver disease: a systematic review and meta-analysis”

PONE-D-20-25326R1

Dear Dr. Wang,

We’re pleased to inform you that your manuscript has been judged scientifically suitable for publication and will be formally accepted for publication once it meets all outstanding technical requirements.

Kind regards,

Michael W Greene, Ph.D.

Academic Editor

PLOS ONE

Additional Editor Comments:

Your initial manuscript submission received a favorable review by an external peer reviewer. You have been responsive to the comments from the Academic Editor. Based on these two factors, your revised manuscript has been recommended for acceptance.

---

## [Editor Report · Acceptance letter]

15 Jan 2021

PONE-D-20-25326R1 

Site-specific risk of colorectal neoplasms in patients with non-alcoholic fatty liver disease: a systematic review and meta-analysis 

Dear Dr. Wang:

I'm pleased to inform you that your manuscript has been deemed suitable for publication in PLOS ONE. Congratulations! Your manuscript is now with our production department. 

Kind regards, 

on behalf of

Dr. Michael W Greene 

Academic Editor

PLOS ONE